# *Aedes aegypti* Strain Subjected to Long-Term Exposure to *Bacillus thuringiensis* svar. *israelensis* Larvicides Displays an Altered Transcriptional Response to Zika Virus Infection

**DOI:** 10.3390/v15010072

**Published:** 2022-12-27

**Authors:** Karine S. Carvalho, Tatiana M. T. Rezende, Tatiany P. Romão, Antônio M. Rezende, Marcos Chiñas, Duschinka R. D. Guedes, Milena Paiva-Cavalcanti, Maria Helena N. L. Silva-Filha

**Affiliations:** 1Aggeu Magalhães Institute, Oswaldo Cruz Foundation, Recife 50670-420, Brazil; 2Center for Genomic Sciences, National Autonomous University of Mexico, Cuernavaca 62210, Mexico

**Keywords:** Bti, *Aedes* control, ZIKV, immune genes, RNA-seq, transcriptome

## Abstract

*Bacillus thuringiensis* svar. *israelensis* (Bti) larvicides are effective in controlling *Aedes aegypti*; however, the effects of long-term exposure need to be properly evaluated. We established an *Ae. aegypti* strain that has been treated with Bti for 30 generations (RecBti) and is still susceptible to Bti, but females exhibited increased susceptibility to Zika virus (ZIKV). This study compared the RecBti strain to a reference strain regarding: first, the relative transcription of selected immune genes in ZIKV-challenged females (F_30_) with increased susceptibility detected in a previous study; then, the whole transcriptomic profile using unchallenged females (F_35_). Among the genes compared by RT-qPCR in the ZIKV-infected and uninfected females from RecBti (F_30_) and the reference strain, *hop*, *domeless*, *relish 1*, *defensin A*, *cecropin D*, and *gambicin* showed a trend of repression in RecBti infected females. The transcriptome of RecBti (F_35_) unchallenged females, compared with a reference strain by RNA-seq, showed a similar profile and only 59 differentially expressed genes were found among 9202 genes analyzed. Our dataset showed that the long-term Bti exposure of the RecBti strain was associated with an alteration of the expression of genes potentially involved in the response to ZIKV infection in challenged females, which is an important feature found under this condition.

## 1. Introduction

*Aedes aegypti* can transmit several arboviruses, such as dengue, chikungunya, and Zika, which have a significant impact on global public health [1]. The control of *Ae. aegypti* populations is considered a global challenge [2,3], and integrated control methods are necessary to achieve a sustainable reduction of such populations [4]. The microbial larvicides based on the entomopathogenic bacteria *Bacillus thuringiensis* svar. *israelensis* (Bti) are one of the most effective tools to fight mosquito and black fly, owing to its high activity and selective action, allied with its environmental safety [5,6,7,8,9]. The mode of action of Bti is based on the presence of an insecticidal crystal, commonly formed by four major protoxins (Cry11Aa, Cry4Ba, Cry4Aa, and Cyt1Aa), that acts in the larvae midgut. Cry and Cyt are pore-forming toxins; Cry specifically binds to midgut receptors, and destroys the epithelial cells via a synergic mode of action with the Cyt toxin [10,11,12,13]. Briefly, larvae ingest the insecticidal crystals that release the protoxins that are further processed into toxins in the midgut. The Cyt1Aa toxin inserts into the membrane of epithelial midgut cells and subsequently binds to Cry toxins, promoting their oligomerization. Cry oligomers then bind with high affinity to the specific receptors on epithelial cells (e.g., aminopeptidases, alkaline phosphatases, and alpha-glucosidases), provoking cell pore formation and toxic effects that result in larval death [14,15]. 

The multiple protoxins found within Bti crystals and their unique mode of action on the larval midgut are the major factors that counteract the selection of resistance, and no consistent report of resistance to such crystals has been documented to date [5,16,17,18,19]. Bti crystal is among the safest larvicides available, although assessment of its long-term utilization on nontarget organisms needs to be refined, as recently reviewed [20,21]. Other aspects of mosquito biology which could potentially be affected by long-term exposure to Bti have not been widely investigated yet. This is particularly important in endemic areas for arboviruses, which require control interventions throughout the year in order to reduce the density of robust *Aedes* populations permanently established in urban areas [22]. To investigate the impact of continuous Bti utilization to control *Ae. aegypti* populations, we established a laboratory strain named RecBti. This strain has been exposed to Bti for 35 generations in order to mirror the conditions of chronic exposure to this larvicide [17]. For this purpose, third-instar larvae from every RecBti generation were treated with a Bti-based larvicide, and their susceptibility was monitored [17]. RecBti larvae from F_1_ to F_30_ were susceptible to Bti and its toxins in comparison with a reference strain [17]. The activity of the detoxifying enzymes found in the RecBti larvae was similar to that of the reference strain, suggesting that continuous Bti exposure did not induce an increase in the metabolism of xenobiotics, as opposed to the exposure to chemical insecticides, which can significantly increase the activity of such enzymes [23,24]. Another study showed that biological parameters, such as fecundity, fertility, pupal weigh, developmental time, sex ratio, and the hematophagic capacity of the RecBti (F_30_) individuals, did not display alterations compared with a reference strain, but females subjected to artificial oral infection with Zika virus (ZIKV) showed increased susceptibility [25]. Higher ZIKV infection rates in the RecBti females were detected at 7 and 14 days post-infection (dpi), compared with RecL. Viral dissemination after 7 dpi was two-fold higher in the RecBti females than in the control females. RecBti also showed higher infection and dissemination with DENV-2, but this was not statistically significant, unlike in the case of ZIKV [25].

Together, these findings showed that the RecBti individuals displayed similar susceptibility to Bti and other biological parameters described above, compared with the reference strain, except for the increased susceptibility of females to ZIKV [25]. We hypothesized that the background of Bti exposure could be associated with the immune response to arbovirus. Therefore, we investigated the profile of expression of specific immune genes in RecBti females challenged with ZIKV, with previously confirmed heightened susceptibility to this virus, as well as the constitutive transcriptome of unchallenged females in comparison with a reference strain.

## 2. Materials and Methods

This investigation is part of a longitudinal study of the RecBti strain, which has been chronically exposed to Bti. Previous findings [17,25] and aspects investigated in the present study are summarized in Figure 1. In this study, the relative expression of seven immune genes in RecBti females from F_30_, which showed an increased ZIKV susceptibility in a previous study [25], was compared with females from the RecL reference strain. Next, the susceptibility of F_35_ larvae to Bti was also assessed by comparing it with the Rockefeller reference strain, as part of the monitoring of this colony during its long-term exposure to Bti [17]. Finally, the whole transcriptome of F_35_ females unchallenged with ZIKV was compared with the RecL strain.

### 2.1. Aedes aegypti Strains

RecBti and the two reference strains used, RecL and Rockefeller, were maintained in the insectarium of Aggeu Magalhães Institute at the Oswaldo Cruz Foundation (IAM-FIOCRUZ). RecBti was established in 2011 from the same geographic area as RecL (Recife, Brazil); at every generation, around 9500 third-instar larvae were treated with a Bti-based larvicide that caused around 74% mortality during the pre-imaginal phase. The adults that survived the Bti treatment during their larval phase were used as the next parental generation. The procedures of Bti exposure were fully described in Carvalho et al. [17]. In this study, RecBti individuals from the F_30_ or F_35_ generations were investigated, as specified in each section. RecL was the reference used, except for the evaluation of Bti susceptibility, which was based on the international reference adopted for this specific purpose, which is the Rockefeller strain. RecL is a local reference strain that has been maintained in the insectary without exposure to any control agent since 1996 [26]. The Rockefeller strain was kindly provided by the Laboratório da Superintendência de Controle de Endemias (SUCEN, Marília-SP) and has been maintained in our insectary since 2007. The Rockefeller strain has been used for monitoring the larvicide susceptibility of RecBti since this strain was established. All three strains were kept under controlled conditions of 26 ± 1 °C, 70% humidity, and a 14 h/10 h light/dark photoperiod. Larvae were reared in dechlorinated tap water and fed with cat food (Friskies^®^). Adults were fed with a sucrose solution (10%) *ad libitum* and females were also artificially fed with defibrinated rabbit blood once per week.

### 2.2. RNA Samples for the Transcription Profile of Immune Genes

The relative expression of immune genes potentially involved in the mosquito’s response to arbovirus infection was investigated using selected RNA samples from a previous study, which showed that F_30_ RecBti females had a significantly increased susceptibility to ZIKV, compared with the reference RecL strain [25]. In summary, those assays investigated nulliparous 7- to 10-day-old females fed on defibrinated rabbit blood containing cultures infected with ZIKV (1:1 ratio), which was previously quantified. Females from untreated control groups, fed blood containing uninfected cell cultures, were used. Infection and dissemination rates were followed up at 3, 7, and 14 days post-infection (dpi). In the present study, we compared two pools of RNA (five female heads with attached salivary glands per pool) of the RecBti strain that showed the highest ZIKV dissemination rate (%) at 7 dpi, with two pools of the RecL females which had fewer disseminated individuals. The following RNA pools from the assays described by Carvalho et al. [25] were evaluated: one pool showed 78.1% of the RecBti-disseminated females versus 33.5% of the RecL-disseminated ones; a second pool from another assay contained 85% of the RecBti-disseminated females versus 50% of the RecL-disseminated ones. Two negative control pools (five female heads with attached salivary glands per pool) were also analyzed: they contained uninfected females from each strain and condition. The relative quantitative transcription of each target gene was then performed using two pools of female heads, tested in duplicate for each strain (RecBti and RecL) and each condition (infected and uninfected). It is worth noting that these F_30_ females were not exposed to Bti during their larval phase, and the last treatment was performed against F_29_ larvae. The transcription profile of seven genes (Appendix A) was investigated, and they were selected on the basis of their potential involvement in immune pathways, as reported in previous studies: *hop* [27], *domeless* [28], *relish* [29], *cactus* [28], *defensing* A [30], *cecropin G* [29] and *gambicin* [31].

### 2.3. RT-qPCR Assays 

Quantitative reverse transcription polymerase chain reaction (RT-qPCR) assays were performed for two purposes: first, to evaluate the transcription of the seven selected genes related to the immune response described in Section 2.2; then, to validate the transcription status of some Differentially Expressed Genes (DEGs) revealed by the RNA-seq described in Section 2.6. The reactions were carried out using a one-step QuantiTect^®^ SYBR Green RT-PCR^®^ Kit (Qiagen, Hilden, Germany), following the manufacturer’s instructions and specific primers for each gene (Appendix A). The ribosomal proteins *rps17* and *18S* were used as endogenous control genes for the assays of the genes involved in the arbovirus response and for selected DEGs shown by RNA-seq, respectively [32]. The samples were analyzed in a QuantStudio^®^ 5 System (Thermo Fisher Scientific, Waltham, MA, USA) and the relative quantification was performed using Applied Biosystems^TM^ Analysis software, Relative Quantification Analysis Module v.3.3 [33,34]. For the immune genes, we first compared the gene expression within each strain fed with ZIKV-infected and uninfected blood using 2^−ΔCtZIKV^/2^−ΔCBlood^. Afterward, a comparison between strains fed with ZIKV-infected and uninfected blood was performed using 2^−ΔCtRecBti^/2^−ΔCRecL^. Then, the means and standard errors from two biological replicates of each strain (RecBti and RecL) and condition (ZIKV-infected and uninfected blood meals) were compared. For the validation of the DEGs, the means and standard errors from three biological replicates of each strain were analyzed. These analyses were carried out using Student’s *t*-tests in GraphPad Prism v.5.0.0 for Windows (GraphPad Software Inc., San Diego, CA, USA), considering a *p*-value < 0.05 to be statistically significant.

### 2.4. Bti Susceptibility Bioassays

The susceptibility of the respective F_35_ larvae to Bti, Cry11Aa, and Cry4Ba protoxins was investigated, prior to the RNA-seq of F_35_ females. For that purpose, bioassays were performed to determine the Lethal Concentrations (LCs) of these compounds for 50% (LC_50_) and 90% (LC_90_) after 24 h, compared with Rockefeller reference larvae [17]. Briefly, groups of twenty late third-instar larvae, in 100 mL tap water in triplicate, were treated with five to seven concentrations that provide a response between 0% and 100% mortality after exposure. A triplicate of untreated groups was run in each test. The assay for each compound and mosquito strain was performed at least three times, and the LCs for each assay were determined through Probit analysis (Statistical Package for the Social Sciences, SPSS 16.0 for Windows). Resistance ratios (RRs) were provided between the LC for the RecBti strain and the respective LC for the reference strain. The toxicity of Bti to the RecL reference strain was reported as additional data.

### 2.5. RNA-seq Library Preparation

To compare the constitutive transcriptomic profile between the RecBti (F_35_) and RecL females, tests were run with RNA samples from three pools of twenty nulliparous 7- to 10-day-old females from each strain. The females from both strains were not blood fed, and the RecBti females were not exposed to Bti during their larval phase. Total RNA samples of whole females from both strains were obtained using the RNeasy Mini Kit™ (Qiagen, Hilden, Germany) following the manufacturer’s instructions. RNA purity and concentration were determined using NanoDrop 2000^TM^ and Qubit 2.0 (Thermo Fisher Scientific). RNA integrity was verified by visualization of samples separated by electrophoresis in 2% agarose gel. Paired-end libraries were prepared from total RNA using a TruSeq™ Stranded mRNA Library Prep Kit (Illumina, San Diego, CA, USA), following standard procedures, and sequenced using a MiSeq Reagent Kit V3™ (Illumina) for 150 cycles on an Illumina MiSeq Sequencer™ (Illumina) at IAM-FIOCRUZ.

### 2.6. RNA-seq Data Analysis 

The quality of the sequenced reads was evaluated by applying the FastQC tool version 0.11.5 (*www.bioinformatics.babraham.ac.uk/projects/fastqc*, accessed on 16 August 2022), and we found that, on average, all the bases had Phred scores higher than 30; thus, the removal of low-quality reads was not needed. Each library was then mapped against the genome assembly of the *Aedes aegypti* strain (Liverpool strain AaegL5) available in the VectorBase database accessed on 12 November 2019 (http://www.vectorbase.org). For the mapping step, the STAR aligner v.2.5.3 [35] was used with default parameters and the quantMode option, which generates GeneCounts considering stranded libraries. The data on the gene counts were then loaded into the R environment and all the replicates were organized in a matrix, which was used as input for the EdgeR package [36]. Significant differential expression was analyzed for each gene using the Quasi-likelihood F-test and multiplicity correction through the Benjamini–Hochberg method to *p*-values, in order to control the false discovery rate (FDR). To perform an exploratory analysis with the data, first, the expression data were used as input for the *plotMDS* function of the EdgeR package. EdgeR was used to perform differential expression analysis, considering only genes with one count per million (CPM) for at least all replicates of a group of samples (reference or RecBti strain). Genes with absolute values of log2 fold change (LFC) ≥1, and with FDR-corrected *p*-values ≤ 0.05, were considered to be DEGs. We selected the DEGs and plotted a scaled heatmap using log2 CPM with the heatmap R package. We created an MA plot that highlighted the DEGs. To further explore and refine the annotation of several DEGs tagged as hypothetical proteins in the *Ae. aegypti* genome, three additional annotation steps were performed using the Blastx tool with default parameters. The mRNA nucleotide sequences from all DEGs were used as queries against the most recent version of the following databases: (i) UniProtKB curated database; (ii) non-redundant database from NCBI, retaining only the top hit for annotation; and (iii) non-redundant database from NCBI using Blast2GO [37], retaining the top five hits for annotation. All databases were downloaded up to March 2020. After that, an additional effort to identify several genes described as hypothetical proteins in the first analysis was made by manual annotation available for *Ae. aegypti* using AnnotationHub version 3.2.2 [38] from the record AH97003. We performed Gene Set Enrichment Analysis (GSEA) using the clusterProfiler R package [39]. Genes were ranked using the LFC. We tested enrichment in the Gene Ontology (GO) and in the Kyoto Encyclopedia of Genes and Genomes (KEGG) gene sets.

## 3. Results

### 3.1. Profile of Immune Gene Transcripts in ZIKV-Challenged RecBti Females (F_30_)

To investigate whether the higher ZIKV dissemination rate in RecBti-infected females from F_30_ revealed in a previous study [25] was related to changes in the expression profile of the genes involved in arbovirus infection, seven targets were compared with a reference strain using pools of RNA extracted from the heads with attached salivary glands of those females. Those target genes were based on their involvement with the immune response to arbovirus previously reported in the literature (Appendix A). First, the *hop* (AAEL012553) and *domeless* (AAEL012471) genes from the JAK-STAT pathway, which can stimulate the antiviral response, were investigated. In the comparison of infected and uninfected females analyzed within strains, both genes were significantly downregulated in RecBti, while they were upregulated in RecL (Figure 2). When infected females from the RecBti and RecL strains were compared, *domeless* and *hop* were significantly downregulated in RecBti (Figure 2). The Toll pathway markers *relish 1* (AAEL007696) and *cactus* (AAEL000709) were also evaluated. The gene *relish 1*, a target that stimulates viral defense, was significantly downregulated after viral infection in both strains, compared with their respective non-infected counterparts. When the status of infected females from the strains was compared, the expression was reduced in the infected RecBti females, although this was not statistically significant (Figure 2). A negative modulator of the toll pathways, *cactus*, showed a reduction in the transcription level in infected individuals within both strains, although this was not significant. The analysis between strains showed an increase in expression in the infected RecBti females, but this was not statistically significant (Figure 2).

Three antimicrobial peptides (AMPs), which can be activated by the IMD or toll pathways after ZIKV infection in mosquitoes [28,40], were also assessed. For *defensin A* (AAEL003841) and *cecropin G* (AAEL015515), no expression was detected in the RecBti-infected individuals (Figure 3). Therefore, the comparison of infected and uninfected conditions, within the RecBti strain and between strains, was not possible, and the full dataset is available in Appendix A. In those assays, the transcription quantification of *defensin A* and *cecropin G*, which was only detected in the RecL strain, showed the downregulation of *defensin A* in the ZIKV-infected females, and there was a reduction between the RecBti and RecL non-infected strains (Figure 3). This suggests that the transcription of *defensin A*, which is potentially involved in the defense to ZIKV, was dramatically repressed in the RecBti individuals (Figure 3). While *cecropin G* was not was affected by ZIKV infection in the RecL strain, it showed a discrete downregulated status in the RecBti females, although this difference was not statistically confirmed (Figure 3). The gene for *gambicin* (AAEL004522) was the only AMP among the peptides tested that displayed detectable expression levels for both strains and conditions. Within strains, *gambicin* was repressed in the infected RecBti females, as well as between infected strains (Figure 3). The results showed that, overall, a trend of repression was detected for all the genes analyzed, except for *cactus*, and the transcription profile of these genes agrees with the status of greater susceptibility to ZIKV shown by the RecBti females.

### 3.2. Susceptibility of RecBti (F_35_) larvae to Bti and its Toxins

Prior to the assessment of the transcriptome using F_35_ females unchallenged with ZIKV described in Section 3.3, susceptibility was determined for the RecBti F_35_ larvae to Bti and its protoxins. This evaluation was part of the continuous monitoring of this phenotype during the maintenance of this strain under Bti exposure. The RecBti F_35_ larvae were susceptible to Bti according to the LC_50_ and LC_90_ values, which displayed only discrete variations, compared with the Rockefeller reference larvae (Table 1). The toxicity of Bti to the RecL strain was assessed and the susceptibility of larvae was similar to the Rockefeller larvae, which was used as the reference for this evaluation. There was no increase in these LCs compared with the last evaluations performed using F_30_ larvae [17]. Likewise, the susceptibility of the RecBti larvae to the Cry11Aa and Cry4Ba protoxins, which were also evaluated as potential markers of selection pressure, was similar to that of the Rockefeller reference strain (Table 1). For the Cry toxins, only the LC_50_ was determined, since individual toxins have low toxicity and LC_90_ is hardly achieved under this condition. The LCs of Bti and both Cry protoxins to RecBti F_35_ larvae displayed a narrow range of variation (RR ≤ 2.2) and confirmed the susceptible status of the larvae to these compounds.

### 3.3. Transcriptomic Profile of RecBti Females (F_35_) by RNA-seq

In this analysis, we sought to determine if exposure of the RecBti *Ae. aegypti* strain to Bti for 34 generations was related to a distinct gene expression profile in RecBti females (F_35_) that were not challenged with ZIKV. These females were compared with females from the reference strain (RecL), obtained under the same conditions, using a next-generation sequencing approach. For this purpose, F_35_ larvae were not treated with Bti and the respective females were not fed on blood. RNA samples obtained using pools of whole females from each strain, in triplicate, were subjected to transcriptome shotgun sequencing (RNA-seq). Analysis of the sequenced RNA-seq libraries showed a total of 10,103,560 and 13,058,581 reads obtained for the RecBti and RecL strains, respectively, and 87% and 85% of these genes were specifically mapped against the *Ae. aegypti* reference genome (Appendix A), respectively. The MDS plot showed a consistent pattern among the replicates for each strain (RecBti: Bti2, Bti3, and Bti4; RecL: Lab3, Lab4, and Lab6), which allowed a comparative analysis (Appendix A). A heatmap of the DEGs generated by the replicates showed a global view of the differential expression observed between the RecBti and RecL strains (Figure 4a). Analysis by the EdgeR package was performed based on 9202 genes (Appendix A), corresponding to around 46% of the total repertoire of genes, which showed at least one CPM in three samples and were also selected using the function *filterByExpr* from EdgeR. 

Among the set of 9202 genes (Appendix A), only 59 Differentially Expressed Genes (DEGs), considering values of LFC ≥ 1 and *p* ≤ 0.05, were found in the RecBti females; 34 were repressed, and 25 were induced (Figure 4b, Table 2). Overall, the constitutive transcription profile of the RecBti females showed a limited set of induced or repressed genes. Description of the DEGs was performed using automatic annotation in the VectorBase database, in addition to manual blast searches against UniProt and GenBank to enhance the identification of genes, since several were detected as hypothetical proteins (Appendix A). Of the 34 downregulated genes, the top gene with LFC of −4.92 was described as a hypothetical protein. The following top DEGs detected with annotation showed LFC values between −2.67 and −1.52, they were: methionine aminopeptidase 1b (−2.67), kielin (−2.35), diacylglycerol kinase 1 (−1.86), poor IMD response (−1.80), lncRNA (−1.78), endothelial PAS domain-containing protein 1 (−1.68), malic enzyme (−1.54), short-chain dehydrogenase (−1.52), and putative adult cuticle protein (−1.51). For the set of upregulated genes, the five most-induced genes exhibited an LFC between 2.14 and 1.83. These included vitellogenic carboxypeptidase precursor (2.14), lnc_RNA (1.92), Mucin-2 (1.90), cathepsin b (1.89), and MRAS-2 (1.83). 

The investigation of the transcription status of some DEGs, which were selected based on their LFC value, adjusted *p*-value, and availability of functional description, was performed by RT-qPCR assays. Seven downregulated genes investigated by RNA-seq, malic enzyme (AAEL005790), adult cuticle (AAEL014978), glutamate decarboxylase (AAEL001902), timeless protein (AAEL019461), and alpha-amylase (AAEL013421), exhibited a profile of reduced expression; however, these were not statistically significant (Figure 5). For diacylglycerol kinase 1 (AAEL019709) and poor Imd response (AAEL021557), the repression status was not confirmed, even if they displayed the highest LFC values in the RNA-seq analysis compared with the five other transcripts (Figure 5). Among the most upregulated genes assessed, vitellogenic carboxypeptidase (AAEL006563) showed significantly higher transcription (Figure 6). For the upregulated cathepsin b (AAEL009642), there was no transcription by RT-qPCR assay. Some top DEGs that were shown in Table 2 were not assessed because when the assays were performed, their description was not available, and they were identified only after a second round of annotation search, as described in Section 2.6. This dataset corroborates the profile of one induced gene in RNA-seq, while this result was not found for the repressed ones. To better exploit this dataset, we also searched for the transcription status of 42 genes, previously reported in the literature as being related to changes potentially associated with the condition of Bti exposure (e.g., genes encoding toxins receptors), and genes related to the immune response or to other insecticide resistance mechanisms (Appendix A). Considering the cutoff used in our study, most of them did not show a differential expression profile in the RecBti individuals. This included the seven immune genes investigated in F_30_ females challenged with ZIKV in this study (Appendix A). Exceptions in this dataset were: the downregulation of an alpha-amylase (AAEL0134210), which was previously reported for its capacity of interaction with Cry toxins; poor Imd response (AAEL021557); leucine-rich immune protein (AAEL0106560); and putative adult cuticle protein (AAEL015424), which can be associated with the immune response and mechanism of insecticide penetration (Table 2). 

To investigate whether specific metabolic processes could be associated with long-term exposure to RecBti, all DEGs were assigned to KEGG pathways. Amongst the 59 DEGs from the EdgeR analysis, 15 were assigned to KEGG pathways, of which seven and eight showed terms specifically enriched for the downregulated and upregulated DEGs of the RecBti strain, respectively. These results of gene set enrichment analysis in RecBti showed only a few changes compared with the reference strain (Appendix A). Most pathways showed only one or two hits; therefore, there was a significant enrichment of these pathways. 

## 4. Discussion 

*Ae. aegypti* RecBti females that displayed an increased susceptibility to ZIKV in previously performed laboratory infection experiments [25] herein showed an altered expression of some genes potentially involved in the arbovirus response. Previous studies have found that these genes are crucial for the defense against ZIKV [27,28,29,30,31]; however, only a few have assessed them in mosquitoes challenged with larvicides [27,41,42], particularly under the condition of long-term exposure, as investigated in our study. We provided evidence that a reduced expression of genes from the JAK-STAT and Toll immune pathways was associated with the increased viral susceptibility found for the RecBti strain subjected to Bti exposure for 29 generations. The data are consistent with the potential activation of the *Ae. aegypti* JAK-STAT pathway in response to viral infection, as reported in previous studies [27,42], including response to ZIKV [28]. The downstream component *domeless* plays an essential role in the regulation of antiviral signaling through the JAK-STAT pathway [43]. In our study, transcripts of the *domeless* and *hop* genes from that pathway showed to be upregulated in the ZIKV-infected reference strain, while they were downregulated in the RecBti strain. Downregulation of these immune genes in ZIKV-infected RecBti individuals was confirmed using a comparison between strains. The Toll pathway plays an important role in *Ae. aegypti* antiviral immune responses [29,30,31,44]. Activation of this pathway in *Ae. aegypti* in response to ZIKV was shown by RNA-seq, and the depletion of cactus, a negative regulator from Toll, resulted in significantly lower intensity of ZIKV infection [28]. Herein, we observed that relish 1, which stimulates the viral defense, showed significant downregulation in ZIKV-infected RecBti females, while this was not found for cactus. Thus, the expression pattern of this target, which is an antagonist in the Toll pathway, suggests the involvement of this pathway for arbovirus defense in the RecBti strain. The activation of five Toll-like receptors from this pathway in *Ae. aegypti* resistant to permethrin, after ZIKV infection, was also recorded [41]. 

Antimicrobial peptides can be produced by induction of the immune signaling pathway in order to neutralize viral pathogens [45]. The Toll and/or IMD pathways, which can be activated in response to viral infection, can trigger the transcriptional activation of defensins and cecropins [40], while gambicin can be induced by the IMD, Toll, and JAK-STAT pathways via combinatorial regulation [46]. In our study, the *defensin A* and the *cecropin G* genes were downregulated in the ZIKV-infected RecL strain. A comparison of uninfected females between strains showed that RecBti females had reduced expression of those AMPs, compared with RecL females, and this difference was significant for *defensin A.* The relative expression of the *defensin A* and *cecropin G* genes between infected strains could not be compared, owing to the undetectable transcription levels in RecBti individuals. Therefore, the data suggest that, in both strains, the expression of these AMPs was downregulated, and in RecBti females, their expression was so low that they were not detected by RT-qPCR. Previous studies have demonstrated the specific defensin A action against ZIKV infection in *Ae. aegypti* [30]. For gambicin, a comparison between infected strains showed reduced expression in RecBti females after viral infection. Therefore, the AMPs investigated herein may be relevant to understand that the immune system can be altered as a function of long-term exposure to Bti and, thus, can trigger a change in the viral susceptibility of these individuals. Together, the quantitative expression of these genes in a pool of females that showed increased ZIKV susceptibility demonstrated a profile of repression. The lack of detection of some AMPs transcripts in RecBti-infected samples is likely to reinforce the repressed transcription profile found for this strain. In this study, the altered expression of immune genes in ZIKV-infected females was investigated in heads with attached salivary glands, which is the last escape barrier for the virus development in mosquito. It is important to consider that other tissues can be assessed, such as the midgut which is the first escape barrier for the virus and also the site of action of Bti toxins. It should be considered that although the gene expression can display variations according to the tissue analyzed [29,47,48,49,50], our data demonstrated a trend of repression of some transcripts involved in arbovirus response from IMD, Toll, and JAK-STAT pathways in the head of infected females, which corroborates their increased susceptibility to ZIKV. However, it is important to note that the antiviral response in mosquitoes can be also regulated by other pathways, whose related genes were not assessed in this study [51].

An important aspect that remains unknown is Bti’s ability to impact larvae and/or adult microbiota. Tetreau et al. [52] showed that *Ae. aegypti* larvae exposed to Bti for 25 h experienced significant changes in the midgut microbiota. Another study showed that variations in the microbiota of larvae can have an impact on the microbiota of adults and their susceptibility to DENV-2 [53]. The role of the microbiota and its impact in mosquito susceptibility to arbovirus has been demonstrated by several studies [54,55,56,57,58]. Therefore, the exposure of mosquitoes to microbial larvicides should be investigated to elucidate their role in the microbiota and the respective impact on the susceptibility to pathogens. It is important to notice that the potential role played by the genes in the respective immune pathways was based on the literature only, as their role was not directly assessed in our study.

In another step of our study, the transcriptome of RecBti females was assessed to offer a broader view of the expression status of genes that could be differentially expressed by RecBti F_35_ unchallenged females, in this case, not induced by ZIKV infection. Before this analysis, the susceptibility of F_35_ larvae to Bti was confirmed, corroborating previous studies which assessed this strain until F_30_ [17,25] and other investigations which showed that long-term exposure to Bti did not evolve to resistance to the Bti crystal [5,16,19]. The transcriptome of RecBti F_35_ females, after 34 generations had been exposed to Bti-based larvicides, but not submitted to ZIKV infection, did not show a marked profile of changes compared with the reference females. RecBti females had a similar profile to the reference strain, as only a limited set of DEGs were detected considering the cutoff used in this study. Genes encoding for Cry toxin receptors, endopeptidases involved in Bti protoxin processing, detoxifying enzymes, immunity, metabolism of xenobiotics, and chitin/cuticle metabolism that showed differential expression through RNA-seq analysis in other *Ae. aegypti* strains exposed to Bti [59,60,61], or in *Ae. aegypti*-resistant strains to individuals Bti toxins [60,62,63], in general, were not found to be differentially expressed in RecBti. It is worth noting that none of the seven altered immune genes found in F_30_ females challenged with ZIKV were found to be differentially expressed in the transcriptome of unchallenged F_35_ females, which is likely to be related to this condition. The transcriptomic profile is in agreement with previous studies which showed that RecBti and the reference strains displayed several similar phenotypes investigated previously; for example, the susceptibility to Bti, susceptibility to chemical insecticides, activity of detoxifying enzymes, and some life traits [17,25]. A completely different scenario was found, for instance, in the transcriptome of a *Cx. quinquefasciatus* strain, which displays a high level of resistance to the Binary (Bin) toxin (RR_50_ > 5000) from *Lysinibacillus sphaericus* larvicides. The Bin-resistant larvae, also unchallenged with this toxin, showed a remarkable constitutive profile of differential expression characterized by 1300 DEGs, including the repression of several genes with LFC values greater than 3, including the gene coding the receptor of the Bin toxin, which is responsible for resistance [64]. 

Although the status of susceptibility to Bti was unaltered for the RecBti strain, which is consistent with the transcriptomic profile found for genes involved in the action of Bti, the increased susceptibility to ZIKV was associated with only two repressed DEGs related to arbovirus response; namely, “poor Imd response” and “leucine-rich immune protein,” and the repressed status of the former was not validated by RT-qPCR. On the contrary, the upregulation of vitellogenin-related genes in RecBti females should be noted, as vitellogenin can play an important role not only in reproduction, but also in immunity, including antibacterial action against *Bacillus thuringiensis* [65,66,67,68]. A comparison of *Cx. quinquefasciatus* populations from Florida, with different susceptibility to West Nile virus (WNV), showed that the most susceptible individuals had greater expression of vitellogenin and bigger ovary size after blood feeding. Such a finding suggests a correlation between ovary development and competence to WNV [69]. It should be noted that such variations in arbovirus susceptibility of mosquito populations have been documented [70,71,72,73,74], but the factors playing a role in such variations remain unknown. In our study on the RecBti strain kept under long-term exposure to Bti, there was an association between the increased susceptibility to ZIKV and the altered transcription profile of genes involved in the immune response to arbovirus. Nevertheless, it should be highlighted that continuous exposure of this *Ae. aegypti* strain to Bti did not impact its susceptibility to this larvicide.

## 5. Conclusions

Long-term exposure of the RecBti strain *Ae. aegypti* to the microbial larvicide Bti did not evolve to an alteration of the susceptibility to this larvicide, but it was associated with a repressed transcription status of immune genes in ZIKV-challenged females, which is consistent with their increased susceptibility to arbovirus. The whole transcriptome of RecBti unchallenged females did not reveal a remarkable set of differentially expressed genes, including most genes related to immune response, under the constitutive condition. Therefore, our dataset indicates the potential influence of long-term exposure to microbial larvicides in the modulation of transcripts potentially involved in the response to ZIKV infection in challenged females, and the mechanisms involved require further investigation, considering the importance of this feature found to be associated with this condition. 

## Figures and Tables

**Figure 1 viruses-15-00072-f001:**
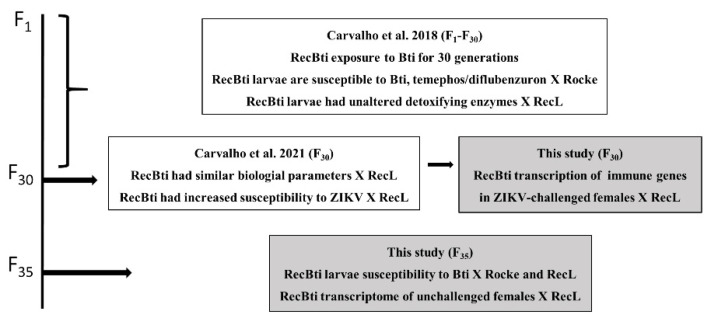
Summary of previous studies and the actual study (gray squares) of the *Aedes aegypti* RecBti strain, chronically exposed to *Bacillus thuringiensis* svar. *israelensis* from F_1_ to F_35_, compared with the reference strains (Rocke, RecL).

**Figure 2 viruses-15-00072-f002:**
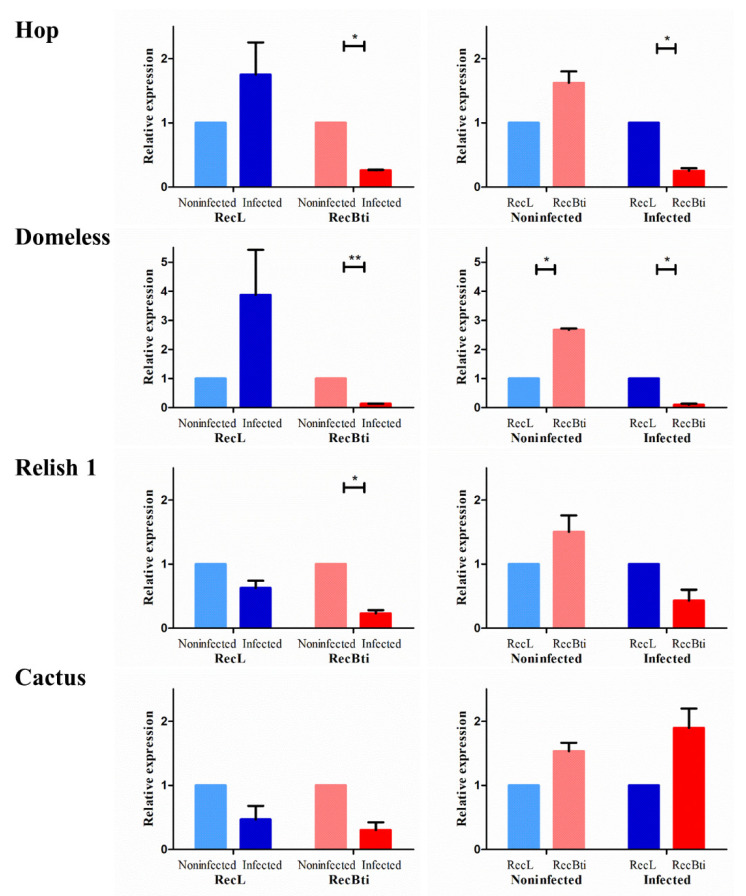
Relative expression of the genes coding proteins from the JAK-STAT (hop and domeless) and toll pathways (relish 1 and cactus) in *Aedes aegypti* from a strain chronically exposed to *Bacillus thuringiensis* svar. *israelensis* (RecBti F_30_) compared with a reference strain (RecL). Comparison of pools (five female heads) from noninfected and infected samples within each strain (left) and between strains (right). Each column and bar represent the average and standard deviation of two assays. Each assay used two pools per condition and strain, in duplicate. Statistical differences of * *p* < 0.05 or ** *p* < 0.005, according to Student’s *t*-tests.

**Figure 3 viruses-15-00072-f003:**
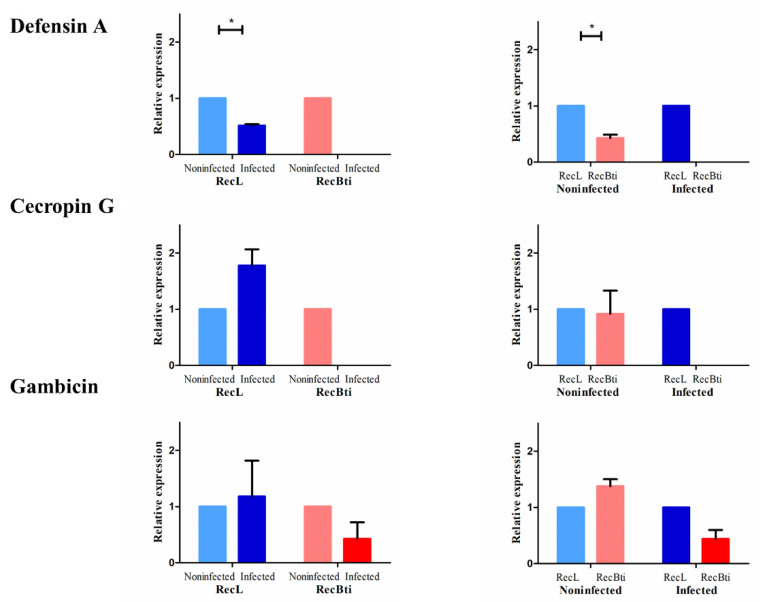
Relative expression of the genes coding antimicrobial peptides (defensin A, cecropin G, and gambicin) from *Aedes aegypti* females from a strain chronically exposed to *Bacillus thuringiensis* svar. *israelensis* (RecBti F_30_) compared with a reference strain (RecL). Comparison of pools (five female heads) from noninfected and infected samples within each strain (left) and between strains (right). Each column and bar represent the average and standard deviation of two assays. Each assay used two pools per condition and strain, in duplicate. Statistical differences of * *p* < 0.05, according to Student’s *t*-tests.

**Figure 4 viruses-15-00072-f004:**
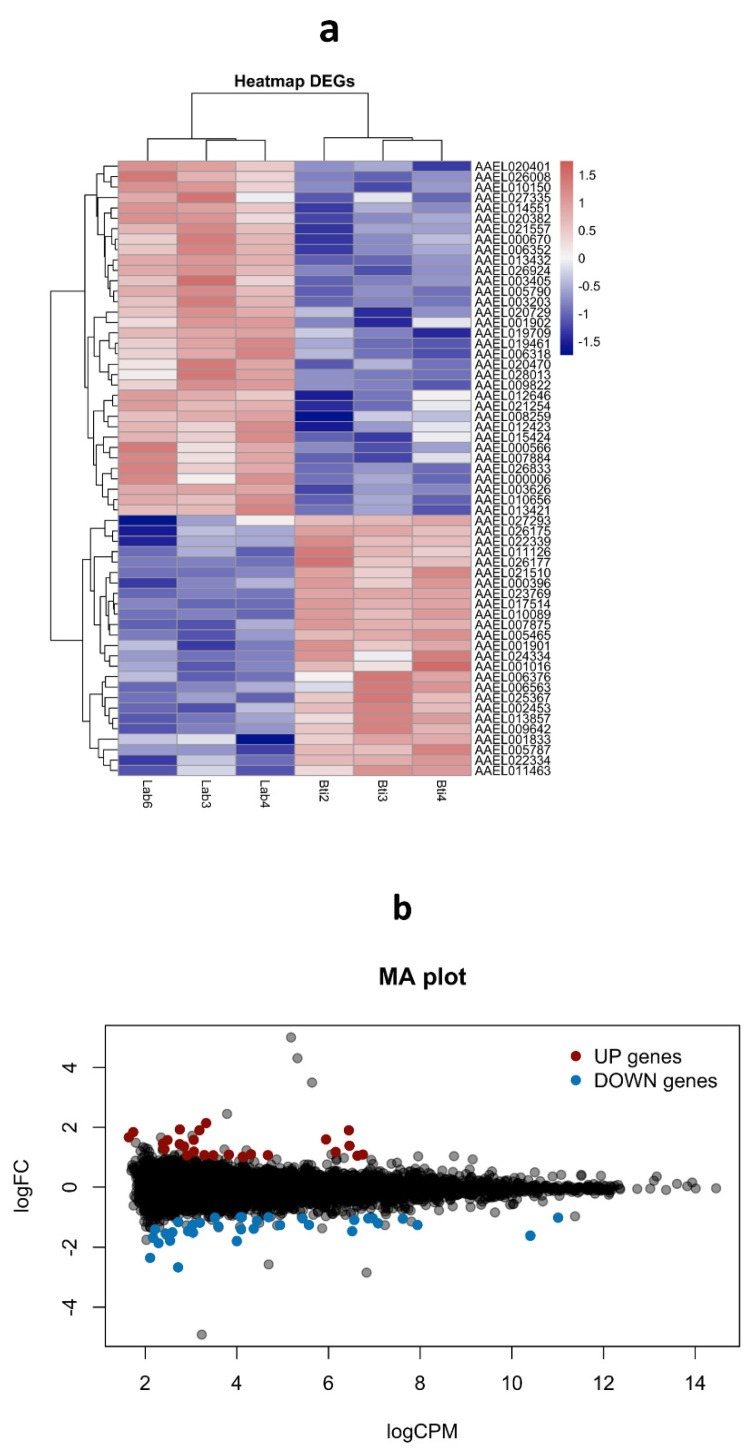
Gene expression profile generated by RNA-seq comparing *Aedes aegypti* females from a strain chronically exposed to *Bacillus thuringiensis* svar. *israelensis* (RecBti F_35_) compared with a reference strain (RecL). (**a**) Heatmap for upregulated (red) and downregulated (blue) genes in biological replicates from the Bti-exposed (Bti2, Bti3, and Bti4) and reference strains (Lab6, Lab3, and Lab4). (**b**) MA plot showing 34 (blue) downregulated and 25 upregulated (red) genes (log2 fold change ≥ 1; FDR-corrected *p*-values ≤ 0.05).

**Figure 5 viruses-15-00072-f005:**
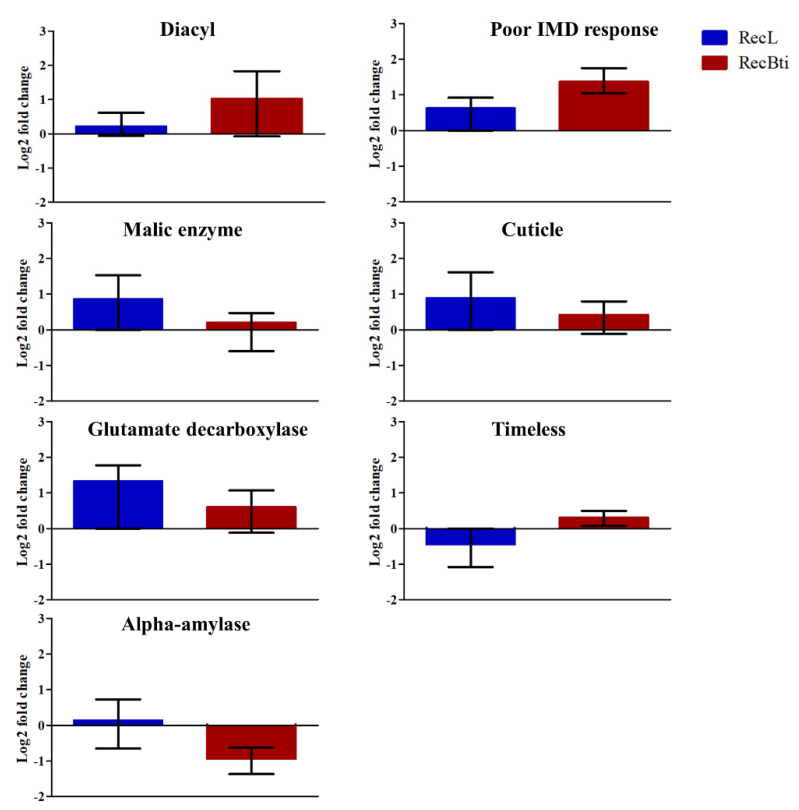
Relative expression of *Aedes aegypti* genes from the RecBti strain chronically exposed to the larvicide *Bacillus thuringiensis* svar. *israelensis* (RecBti F_35_) compared with the RecL reference strain. These were downregulated genes according to RNA-seq analysis from Table 2. Columns represent the average and standard deviation of three RNA samples analyzed in duplicate. No statistical difference was found according to Student’s *t*-tests.

**Figure 6 viruses-15-00072-f006:**
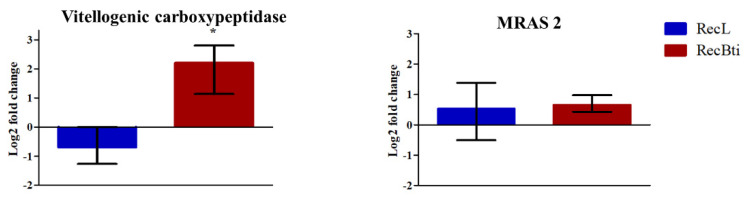
Relative expression of *Aedes aegypti* genes from the RecBti strain chronically exposed to the larvicide *Bacillus thuringiensis* svar. *israelensis* (RecBti F_35_) compared with the RecL reference strain. These were upregulated genes according to the RNA-seq analysis from Table 2. Columns represent average and standard deviation of three RNA samples analyzed in duplicate. Statistical differences of * *p* < 0.05 according to Student’s *t*-tests.

**Table 1 viruses-15-00072-t001:** Toxicity of *Bacillus thuringiensis* svar. *israelensis* (IPS82) and its protoxins (Cry11Aa and Cry4Aa) to *Aedes aegypti* larvae from a strain chronically exposed to Bti for 35 generations (RecBti-F_35_) compared with the Rockefeller reference strain (Rocke) and the RecL strain.

Sample	N	LC_50_ (95% CI) ^a^	R ^b^	LC_90_ (95% CI)	R
Bti					
Rocke ^c^	1620	0.008 (0.007–0.009)	1.0	0.026 (0.021–0.036)	1.0
RecL ^d^	1200	0.009 (0.007–0.010)	1.1	0.027 (0.022–0.040)	1.0
RecBti F_35_	1680	0.015 (0.012–0.013)	1.9	0.030 (0.029–0.039)	1.2
Cry11Aa					
Rocke	1260	0.436 (0.338–0.666)	1.0	ND ^e^	ND
RecBti F_35_	1680	0.717 (0.493–0.968)	1.6	ND	ND
Cry4Ba					
Rocke	1080	0.331 (0.209–0.492)	1.0	ND	ND
RecBti F_35_	1680	0.731 (0.6285–1.24)	2.2	ND	ND

^a^ Lethal concentration (mg/l) for 50% and 90% of third-instar larvae exposed for 24 h (mean and 95% confidential intervals-CI). ^b^ Ratio between the LC of the RecBti and the Rocke strain. ^c^ LCs of the Rocke strain were obtained from Carvalho et al. 2018. ^d^ Data from the Reference Service for Culicide Vector Control from IAM-FIOCRUZ (report RF-AT-003/2021); ^e^ Not Determined.

**Table 2 viruses-15-00072-t002:** Topmost downregulated and upregulated genes in *Aedes aegypti* females from a strain chronically exposed to *Bacillus thuringiensis* svar. *israelensis* (RecBti F_35_) compared with a reference strain (RecL), revealed by RNA-seq and analyzed by EdgeR.

Identity	Description	LFC ^a^	*p*-Value
Downregulated			
AAEL027968	Hypothetical protein	−4.92	2.51 × 10^−2^
AAEL000566	Methionine aminopeptidase 1b	−2.67	2.49 × 10^−5^
AAEL008259	Kielin/chordin-like protein	−2.35	1.56 × 10^−4^
AAEL019709	Diacylglycerol kinase 1	−1.86	4.16 × 10^−4^
AAEL021557	Poor Imd response	−1.80	5.99 × 10^−7^
AAEL026924	lncRNA	−1.78	3.37 × 10^−5^
AAEL020401	Endothelial PAS domain-containing protein 1	−1.68	3.56 × 10^−4^
AAEL005790	Malic enzyme	−1.54	2.14 × 10^−9^
AAEL006318	Short-chain dehydrogenase	−1.52	4.08 × 10^−4^
AAEL015424	Putative adult cuticle protein	−1.51	1.62 × 10^−4^
AAEL001902	Glutamate decarboxylase	−1.47	6.65 × 10^−4^
AAEL019461	Protein timeless	−1.41	2.34 × 10^−7^
AAEL013421	Alpha-amylase	−1.41	5.53 × 10^−5^
AAEL003626	Sodium dependent amino acid transporter	−1.39	5.06 × 10^−7^
AAEL009822	GPCR Metabotropic glutamate	−1.37	1.17 × 10^−3^
AAEL012646	Protein obstructor-E	−1.36	3.52 × 10^−4^
AAEL010150	SH2 domain-containing protein 2A	−1.26	1.86 × 10^−5^
AAEL013432	Serine protease	−1.26	7.07 × 10^−8^
AAEL020382	Acetyl-coenzyme A synthetase	−1.26	1.94 × 10^−6^
AAEL010656	Leucine-rich immune protein	−1.21	3.70 × 10^−8^
AAEL007884	Conserved membrane protein 44E	−1.19	4.51 × 10^−4^
AAEL028013	Alanine-glyoxylate aminotransferase	−1.16	1.42 × 10^−4^
AAEL006352	Hybrid signal transduction histidine kinase B	−1.15	1.86 × 10^−4^
AAEL000006	Phosphoenolpyruvate carboxykinase	−1.05	2.50 × 10^−5^
AAEL026008	Biotin synthase	−1.04	5.27 × 10^−7^
AAEL020729	Ribosomal protein VAR1, mitochondrial	−1.04	1.30 × 10^−6^
AAEL000670	Methionine sulfoxide reductase	−1.03	1.28 × 10^−5^
AAEL014551	Triacylglycerol lipase, pancreatic	−1.03	7.74 × 10^−7^
AAEL003405	Regulator of microtubule dynamics protein 1	−1.02	6.87 × 10^−6^
AAEL003203	Fatty acid desaturase	−1.01	8.69 × 10^−7^
AAEL020470	Serine protease snake	−1.00	5.02 × 10^−4^
AAEL026833	ETS domain-containing transcription fact. ERF	−1.00	9.64 × 10^−6^
AAEL012423	Cell wall integrity stress response component 1	−1.00	2.73 × 10^−4^
AAEL027335	Acyl-CoA Delta-9 desaturase	−1.00	5.63 × 10^−4^
Upregulated			
AAEL006563	Vitellogenic carboxypeptidase precursor	2.14	4.94 × 10^−2^
AAEL024334	lnc_RNA	1.92	5.09 × 10^−3^
AAEL021510	Mucin-2	1.90	4.00 × 10^−4^
AAEL009642	Cathepsin b	1.89	6.55 × 10^−4^
AAEL001901	MRAS2	1.83	3.32 × 10^−2^
AAEL005465	Parkin coregulated gene protein homolog	1.66	4.94 × 10^−2^
AAEL017514	Histidine-rich glycoprotein-like	1.59	3.37 × 10^−11^
AAEL027293	CAAX prenyl protease 1 homolog	1.58	2.70 × 10^−4^
AAEL022339	lnc_RNA	1.57	1,62 × 10^−4^
AAEL006376	Trypsin	1.45	7.61 × 10^−4^
AAEL005787	Serine protease	1.43	9.40 × 10^−5^
AAEL013857	Serine protease Hayan-like, transcript v X2	1.38	1.63 × 10^−6^
AAEL010089	Protein AAR2 homolog	1.36	4.81 × 10^−5^
AAEL023769	40S ribosomal protein S21	1.26	4.86 × 10^−4^
AAEL026177	Acyl-CoA-binding domain ^b^	1.18	1.72 × 10^−4^
AAEL025367	lnc_RNA	1.17	4.55 × 10^−7^
AAEL001833	Juvenile hormone-inducible protein	1.09	2.63 × 10^−4^
AAEL022334	lnc_RNA	1.08	2.10 × 10^−7^
AAEL007875	Leucine-rich repeat ^c^	1.08	2.07 × 10^−5^
AAEL000396	Calcium-dependent protein kinase 1	1.07	1.92 × 10^−4^
AAEL011126	Alcohol dehydrogenase	1.06	7.24 × 10^−6^
AAEL001016	Zinc finger protein	1.06	3.11 × 10^−4^
AAEL011463	Cytochrome P450	1.05	9.04 × 10^−4^
AAEL026175	Apolipophorins-like	1.04	1.05 × 10^−5^
AAEL002453	Zinc finger protein	1.01	3.62 × 10^−5^

^a^ Log2 fold change. ^b^ Acyl-CoA-binding domain-containing protein. ^c^ Leucine-rich repeat and immunoglobulin-like domain-containing nogo receptor-interacting protein 1.

## Data Availability

Raw data set is available in Appendix A; the transcriptome dataset was submitted to NCBI (http://www.ncbi.nlm.nih.gov/bioproject/870084) on 16 August 2022 and will be available under publication.

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
