# Peer review of "Aedes aegypti* Strain Subjected to Long-Term Exposure to *Bacillus thuringiensis* svar. *israelensis* Larvicides Displays an Altered Transcriptional Response to Zika Virus Infection"

_viruses, 2022, doi:10.3390/v15010072_

Round 1
Reviewer 1 Report (Previous Reviewer 3)
The authors have appropriately addressed my comments. I don't have any issues.
Author Response
ANSWERS TO AUTHORS
Viruses Resubmission ID: viruses-2052671
Reviewer 1
Comment. English language and style are fine/minor spell check required.
Answer. The manuscript was revised again by an English editing service.
Other revised aspects:
- In the methodology, we added information that head with attached salivary glands were used (lines 124, 131) as well as in results (line 226).
-The conclusion was improved according to the data obtained in this study.
-Other minor changes were introduced thought the manuscript.

Reviewer 2 Report (Previous Reviewer 2)
The authors have not sufficiently addressed 2 important comments:
Specifically, the authors have not addressed that the midgut is the barrier where the Bti has its effect, and responses may differ depending on the site of replication in the mosquito? Therefore the head or whole females will not be relevant.
RecL toxicity data was not included in table 1. While I understand, the Rockefeller strain is the gold standard, the RecL should still be included (and compared with the gold standard).
Author Response
ANSWERS TO AUTHORS
Viruses Resubmission ID: viruses-2052671
Reviewer 2
Comment. English language and style are fine/minor spell check required.
Answer. The manuscript was revised by an English editing service. Several changes were also introduced to improve the manuscript and those changes were tracked.
Reviewer comment. RecL toxicity data was not included in table 1. While I understand, the Rockefeller strain is the gold standard, the RecL should still be included (and compared with the gold standard).
Answer. As requested, we included the data of Bti toxicity against the RecL reference strain in Table 1. As previously informed in the “Answer to reviewers”, the data now introduced in Table 1 demonstrated that the Bti LC’s towards RecL are similar to those found to Rockefeller strain. Data was kindly provided by the SRCV from IAM-FIOCRUZ (report RF-AT-003/2021). Data derived from three bioassays (n=1200) which showed the following lethal concentrations: LC50= 0.009 (0.007-0.010) and LC90= 0.027 (0.022-0.040). A sentence concerning the similar susceptibility of these susceptible reference strains was included in Results (line 284).
Reviewer comment. Specifically, the authors have not addressed that the midgut is the barrier where the Bti has its effect, and responses may differ depending on the site of replication in the mosquito? Therefore the head or whole females will not be relevant.
Answer. Concerning this specific question, it is important to clarify that Bti is an entomopathogen whose larvicidal toxins (Cry and Cyt) act on the midgut by binding to specific receptors on the epithelium, as described in the introduction (lines 33-36, refs 10-13). Therefore, it should be noted that the midgut is the site of action of Bti toxins, while midgut is considered the first barrier for arbovirus infection. Concerning the choice of tissue sample in our study, we used the heads (with attached salivary glands) for logistical matters and also supported by other studies that investigated immune related-gene expression in this tissue, as the head contains the salivary glands which is last escape barrier for the virus development in mosquito (Luplertlop et al. 2011, https://doi.org/10.1371/journal.ppat.1001252; Chowdhury et al. 2020 https://doi.org/10.1371/journal.ppat.1008754; Chowdhury et al. 2021 https://doi.org/10.1371/journal.ppat.1001252). We had positive results showing alteration of immune genes expression using these samples. The work of Hixson et al. 2021 (doi 10.7554/eLife.76132), in particular, provided an interesting overview of gene expression in different tissues of Ae. aegypti through RNA-seq. This study showed the expression of several immune pathway related genes in different tissues and, for the majority of those genes (including those that we investigated), the expression in heads was more important than in the midgut. For this group of genes the greatest expression was in the ovaries. As suggested by the reviewer, we agree that the midgut is a tissue that could be investigated, as this is the first insect’s escape barrier to viral infection and also the initial site of action of Bti. Nevertheless, it is important to note that these ZIKV-challenged females were not treated with Bti during their larvae phase. Therefore, the altered gene expression detected was not associated to a direct effect of a Bti treatment, but rather with a condition of long-term exposure. Regarding this aspect, we introduced in the manuscript the comment below to reinforce the tissue we used and to clarify that variations can be found among studies:
Line 443. “In this study, the altered expression of immune genes in ZIKV-infected females was investigated in heads with attached salivary glands, which is the last escape barrier for the virus development in mosquito. It is important to consider that other tissues can be assessed, as the midgut which is the first escape barrier for the virus infection and also the site of action Bti toxins. It should be considered that although the gene expression can display variations according to the tissue analysed (Luplertlop et al 2011; Sim et al 2012; Chowdhury 2020, 2021; Hixson et al 2022), our data demonstrated a trend of repression of some transcripts involved in arbovirus response in the head of infected females which corroborates their increased susceptibility to ZIKV. However, it is important to remind that the antiviral response in mosquito can be also regulated by other pathways whose related-genes were not assessed in this study (Tikhe and Dimopoulos 2021 (doi 10.1016/j.dci.2020.103964).)
Other revised aspects:
- In the methodology, we added information that head with attached salivary glands were used (lines 124, 131) as well as in results (line 226).
-The conclusion was improved according to the data obtained in this study.
-Other minor changes were introduced thought the manuscript.

Round 2
Reviewer 2 Report (Previous Reviewer 2)
authors have sufficiently addressed my comments
This manuscript is a resubmission of an earlier submission. The following is a list of the peer review reports and author responses from that submission.
Round 1
Reviewer 1 Report
The manuscript describes the comparison of the transcriptional profiles of two Aedes aegypti strains: one exposed to Bti for 34 generations (RecBti F35) and a reference strain (colony from mosquito’s collected from the same geographical region and without exposition to Bti, RecL), both infected with Zika virus and non-infected. Susceptibility to Bti and related toxins (by comparing RecBti F35 strain with Rockefeller reference strain), transcriptional profile of mosquito immune genes (by comparing RecBti F30 strain with RecL reference strain) and differentially expressed genes (by comparing RecBti F35 strain with RecL reference strain) are also screened.
This work follows previous studies of part of the involved research team with the Aedes aegypti strain subjected to Bti larvicide for 29 generations (strain RecBti F30), presented in references 17 and 23. In this manuscript, this strain is used for some of the experiments and further exposed to Bti for five additional generations (strain RecBti F35).
The manuscript provides substantial data and is soundness regarding the importance of Bti in Aedes aegypti control and related mosquito-borne diseases transmission. However, the text is sometimes confusing, mixing data from the previous studies (references 17 and 23). As it is, the reading of this manuscript, it is difficult to follow, especially if without the access to the previous works. Moreover, the use of two references strains (RecL and Rockefeller, only used to study Bti and related toxins susceptibility [as far as I understood]) and two RecBti strains (F30 and F35) is not always clear throughout the text. I would suggest to add a graphical figure presenting the layout of the experimental design to help to reader to follow the work (and linking to the previous works, to clarify what is newly presented) without the immediate need to seek additional information.
A thorough revision to increase clarification and reading fluidity would be beneficial.
Specifically, I have the following comments and suggestions:
· Title: In the title, please add a dot to svar. or use the full term serovar. Also, this manuscript only present data of transcriptional response to ZIKV infection, “… displays an altered arbovirus transcriptional response” is an overstatement, and should be clarified by substituting arbovirus by Zika virus.
· Authors and Affiliation: please correct to the same letter size for all authors and affilitions.
· Abstract: Rewrite the Abstract to clarify the new data presented and the data/strains reported in previous studies.
· Keywords: I would suggest to remove dengue virus from the keywords or to substitute by Aedes aegypti. There is no new data/experiments performed with dengue virus in the manuscript.
· Introduction: 1st line – Correct Aedes aegypti letter size
· Introduction: end of page 2 –please state the reference strain used in reference 23 for the study of biological parameters. RecL, right? In the last sentence RecL strain is referred without previous clarification that it is the reference strain, used.
· Introduction: last paragraph, page 3 - it is stated that an increased susceptibility to the arbovirus tested (reference 23) but the only validated results were the ones associated to Zika virus. Please clarify, stating the conclusions of reference 23, here only ZIKV is referred.
· Materials and Methods: section 2.5, page 5- “The females from both strains were not fed blood,…” please correct to “blood fed” or “fed with blood”
· Materials and Methods: end of section 2.6, page 6 – I found no previous reference to GO and KEGG gene set before. Please clarify the use of abbreviations/gene references on the first use
· Table 1 – ND the uppercase for footnote should be d, not c. Please correct.
· Figure 3 – Please increase the Heatmap DEGs figure (a) or at least the letter size of the down column (biological replicates and reference strains). It is very difficult to see as presented.
Reviewer 2 Report
The manuscript ‘Aedes aegypti strain subjected to long-term exposure to Bacillus thuringiensis svar israelensis larvicides displays and altered arbovirus transcriptional response’ by Carvalho et al. describes the differences in the transcriptome of A. aegypti exposed to Bti, and what effect on the response to ZIKV infection is. Overall, the study shows only limited differences in responses, mostly guided by specific selection of genes and pathways, rather than a clear differences in the overall transcriptome of treated mosquitoes. Controls to attribute the gene expression to an observed phenotype are lacking as detailed below. These results may be too preliminary as the overall conclusions are too broad and not supported by the data in it’s current form.
General comments:
- The title seems to be misleading as it suggests a general altered response to arbovirus infection, while this was only tested for ZIKV. In fact, from the previous study (ref 23), the potential enhanced infection following exposure to Bti was only observed for ZIKV, not DENV. Therefore, in general it would have been important to include DENV data in this study (figure 1 and 2) to show whether the observed phenotype actually correlates with gene expression findings.
- Some sentences are too long and unnecessary commas are used. For example in the introduction the sentence ‘Therefore, to investigate the impact of … exposure to this larvicide [17].’ Please improve readability by getting rid of too many commas in one sentence or splitting up one sentence into two (throughout the whole manuscript).
Specific comments:
2. Material and Methods
2.1.
- Aedes aegypti strains: What is the rationale behind using Rockefeller as a reference strain if you already have RecL as the reference strain?
- What generations were used for the controls and were they maintained in the same way throughout these generations?
- From the text it seems like RecBti and RecL were not initiated from the same colony? Why not? This would have been a better comparison.
- What is the rationale for using F30 for some, and F35 for other experiments?
- Was RecL mock treated in paralel for 30-35 generations?
2.3.
- what was the rationale for using the heads for immune gene profiling? While the heads could be used as a surrogate for determining dissemination (although legs and wings are generally used for this), the midgut is the barrier where the Bti has its effect, and responses may differ depending on the site of replication in the mosquito?
2.5.
- please detail what tissue was used for this analysis, not clear whether these are also heads, if so, please see my previous comment.
3. Results
3.1.
- please include RecL in the toxicity data from table 1.
3.2.
First sentence: “... was related TO changes in the expression...
‘… which stimulate the viral response, …’ do you mean antiviral response?
Page 8: ‘Therefore, a comparison of infected and uninfected conditions … was not possible due to a lack of transcription in the RecBti samples.’ I don’t understand this cause-effect argumentation. I mean it is a bit more clear in Figure 2 but I still don’t understand why you can say that you do detect the control genes but not the target genes and therefore cannot compare between infected and non-infected. What is the point of doing the assay then or why is a 0 value unusable? Also, for the Defensin A panel there is a value for the infected RecL in the left panel but why not on the right? To me, this whole part/figure is confusing.
Page 12: ‘… or some were assessed but the RT-qPCR assays did not provide robust results’ maybe delete this sentence?
3.4.
- please provide a rationale for the selection criteria of genes in figure 4. Several genes in the top hit list were not tested (Methionine, Endothelial PAS, etc).
4.
- In the discussion it is suggested that changes in the JAK-STAT pathway, among other, play a role in the enhance dissemination of ZIKV. However, it would be important to discuss (or better yet test, see comments above) the differential effect for DENV.
- a comparison of uninfected female between strains showed reduced expression of AMP’s however this was not seen in the transcriptome analyses by RNAseq? Since RNAseq is also a quantitative method for gene expression, these data do not support eachother within this study. One difference is the F30 vs F35, but this is not likely to have such a difference? It would be good to repeat figure 1 and 2 for RNA from F35 used in transcriptomic analysis.
Reviewer 3 Report
Carvalho et al, have evaluated the transcriptional response of Bti treated mosquitoes. Even though, the results can be interesting, there are several short comings. Here are my comments:
1) Sample size, number of replicates and experimental numbers are very low.
2) They measured at a time point (7 days infection), which may not provide a full pictures of gene expression profile. What about early or later time points post-infection.
3) They compared expression profiles based on one phenotype (high zika transmissible mosquitoes in Bti group) and failed to compare the other phenotype (low zika transmissible Bti group). This could also give a better picture of the expression profile.
4) Validation was only based on RT-qPCR. However, they failed to perform the other downstream validation such as RNAi silencing studies to support their results and conclusion.
Minor revisions:
Section 3.2
Can you please include the basis for selection of seven targets? Or provide reference?
Can you provide reference for the following statement?
Three antimicrobial peptides (AMPs), which can be activated by the IMD or Toll pathways after ZIKV infection in mosquitoes, were also assessed
Section 2.3
What was the method used for infecting mosquitoes?
Figure 2: is the right side of panel correct?